# Geometric entropy of plant leaves: A measure of morphological complexity

**Vishnu Muraleedharan**[1,2,3], **Sajeev C. Rajan**[1,3], **Jaishanker R**[1,3,4]*

**1** C V Raman Laboratory of Ecological Informatics, Indian Institute of Information Technology and Management—Kerala, Trivandrum, Kerala, India, **2** Cochin University of Science and Technology, Cochin, Kerala, India, **3** Kerala University of Digital Sciences, Innovation and Technology, Technopark Phase—IV, Thiruvananthapuram, Kerala, India, **4** School of Ecology and Environment Studies, Nalanda University, Rajgir, Bihar, India

* jrnair@duk.ac.in

## Abstract

Shape is an objective characteristic of an object. A boundary separates a physical object from its surroundings. It defines the shape and regulates energy flux into and from an object. Visual perception of a definite shape (geometry) of physical objects is an abstraction. While the perceived geometry at an object's sharp interface (macro) creates a Euclidian illusion of actual shape, the notion of diffuse interfaces (micro) allows an understanding of the realistic form of objects. Here, we formulate a dimensionless geometric entropy of plant leaves ($S_L$) by a 2-D description of a phase-field function. We applied this method to 112 tropical plant leaf images. $S_L$ was estimated from the leaf perimeter ($P$) and leaf area ($A$). It correlates positively with a fractal dimensional measure of leaf complexity, viz., segmental fractal complexity. Leaves with a higher $P$: $A$ ratio have higher $S_L$ and possess complex morphology. The univariate cluster analysis of $S_L$ reveals the taxonomic relationship among the leaf shapes at the genus level. An increase in $S_L$ of plant leaves could be an evolutionary strategy. The results of morphological complexity presented in this paper will trigger discussion on the causal links between leaf adaptive stability/efficiency and complexity. We present $S_L$ as a derived plant trait to describe plant leaf complexity and adaptive stability. Integrating $S_L$ into other leaf physiological measures will help to understand the dynamics of energy flow between plants and their environment.

## Introduction

Nature has invested heavily in diversity. It manifests along multiple dimensions (phenotypic, physiological) and tiers (molecular, individual, population, community). Physical appearance (morphology) is the most obvious trait that can be used to differentiate higher forms of life. Leaf shape of Angiosperms (flowering plants) is an easily discernable plant trait widely used by taxonomists to characterize plant species [1, 2]. The different shapes of leaves have evolved through natural selection. These abstract forms (shapes) are not random. They have a mathematical soul that defines their physical form. Fibonacci numbers [3], the golden ratio [4], and

**Data Availability Statement:** All relevant data are within the paper and its supporting information files.

**Funding:** The author(s) received no specific funding for this work.

**Competing interests:** The authors have declared that no competing interests exist.

fractals [5] provide mathematical descriptions of how biomass is arranged in numerical and structured geometric forms.

Mass and shape are fundamental attributes of living organisms. Biomass limited by a boundary bestows shape to organisms. Living objects exchange energy and matter with surroundings across boundaries. The exchange of energy and matter underpins evolutionary [6] and ecological processes [7]. Mereotopology is a theory combining mereology and topology. It deals with relations of things: parts, wholes, and their boundaries [8]. Mereotopology qualitatively describes the static relationship between neighboring things by logical expressions, either "true (1)" or "false (0)".

The phase-field concept rooted in mathematical physics also describes the boundaries of neighboring things [9]. In contrast to mereotopology, the phase-field concept quantitatively describes geometric shapes, boundaries, and their dynamic evolution (transitions between object and boundary) [10]. Modeling using the phase-field approach belongs to the class involving phase transitions between states. Description of the nature or shape of the transition region of the phase-field function is achieved by the statistical distribution of gradients in the transition region [11]. The concept was first implemented in describing the evolution of complex dendritic structures [12] and later gained the attention of the material science fraternity. Nowadays, phase-field models are widely used to describe complex shapes, boundaries, and evolution [13, 14].

In-depth knowledge of shape and size is a prerequisite to understanding the interaction between objects and their environment. Shape perception approaches focus exclusively on geometric and computational tools. The shape of any physical object is a perception rendered by human vision. Markosian et al. [15] suggest that only objects with spatial locations be considered physical objects. However, spatial bounds alone do not convey information (here, shape) of a physical object. Also, relying on visual perception to conceive the idea of a boundary is unscientific. Visual demarcation at the sharp macroscale interface of an object returns an illusion. Such shapes demarcated as object boundaries will change with magnification. A realistic boundary can be perceived as diffuse microscale interfaces with finite thickness [16–18]. Here, we present a case study in complexity and use the notion of microscale interfaces and phase field transitions to arrive at the geometric entropy of plant leaf shapes.

Plant leaves exemplify remarkable complexity. Leaves are the primary sites that regulate photosynthesis and energy transfer. The amount of solar radiation incident on leaves depends on the geometry and inclination of individual leaves [19]. Converting complex leaf forms into simple geometric shapes can give valuable insights into leaf-radiation interaction. It allows an in-depth study of geometry and energy by the basic properties of Euclidean shapes. Information and entropy unify the idea of geometry and energy in all biological systems [20, 21]. Understanding the joint descriptions of information, entropy, and geometry will open discussions on the direct causal links between leaf stability/efficiency and the complexity of plant leaves. Until the dimensionless form of Bekenstein—Hawking entropy was constructed using a phase-field approach [22], only the notion of entropy in the 'herogeneity' sense [23] was related to the perception of geometry. In this paper, we consider plant leaves to be made up of 2-D non-linear elements and derive their geometric entropy through mathematical formulations based on the geometric approach of Schmitz [22].

## Leaf as a 2-D geometric object

Like any physical object, a plant leaf has a mass, momentum, and temperature. We attempt to describe the geometric entropy of an individual plant leaf by the mathematical information of physical objects gathered from the formulation of the Bekenstein-Hawking entropy [22]. The formulation is purely geometrical and is devoid of any of the attributes of physical objects.

The mean thickness and laminar length/width of plant leaves are in the order of $10^{-4}$ $m$ and $10^{-2}$ $m$, respectively [24]. Since the thickness of the leaves is relatively smaller than their laminar dimensions, the visual perception of leaf morphological features is more laminar. Thus, leaves exhibit remarkable complexity more by variations at their margin, leading to lobes or serrations. 2- D leaf shape features described along the laminar direction can be considered an excellent candidate to discriminate the leaves from others. Hence, we consider the plant leaf as a 2-D structure for analytical purposes. A 2-D leaf is described by an area confined by a boundary—the leaf margin. The leaf boundary distinguishes the bulk of the leaf from its surrounding environment.

As stated earlier, we consider the leaf as a 2-D object made of non-linear elements. Circular objects can be described using a continuous 2-D extension of Heaviside and phase-field functions [25, 26], thereby constructing the geometric entropy at its diffuse interfaces.

## Description of a circular object

The Heaviside step function ($H(x)$) and the Phase-field function ($\Phi(x)$) are essential in describing the physical states (presence or absence) of a system and are used in the modeling and mechanics of complex structures [27, 28]. The sharp interface property of the $H(x)$ yields the value '1' wherever the object is present and is '0' elsewhere (Fig 1). The geometry of a circle can be described using this property. Boundaries distinguish an object ('1') from its non-object state ('0'). A circle is an area confined by its periphery (boundary). We consider the boundary of the circle as a sharp interface. Beyond their boundary, there will be no presence of the circle.

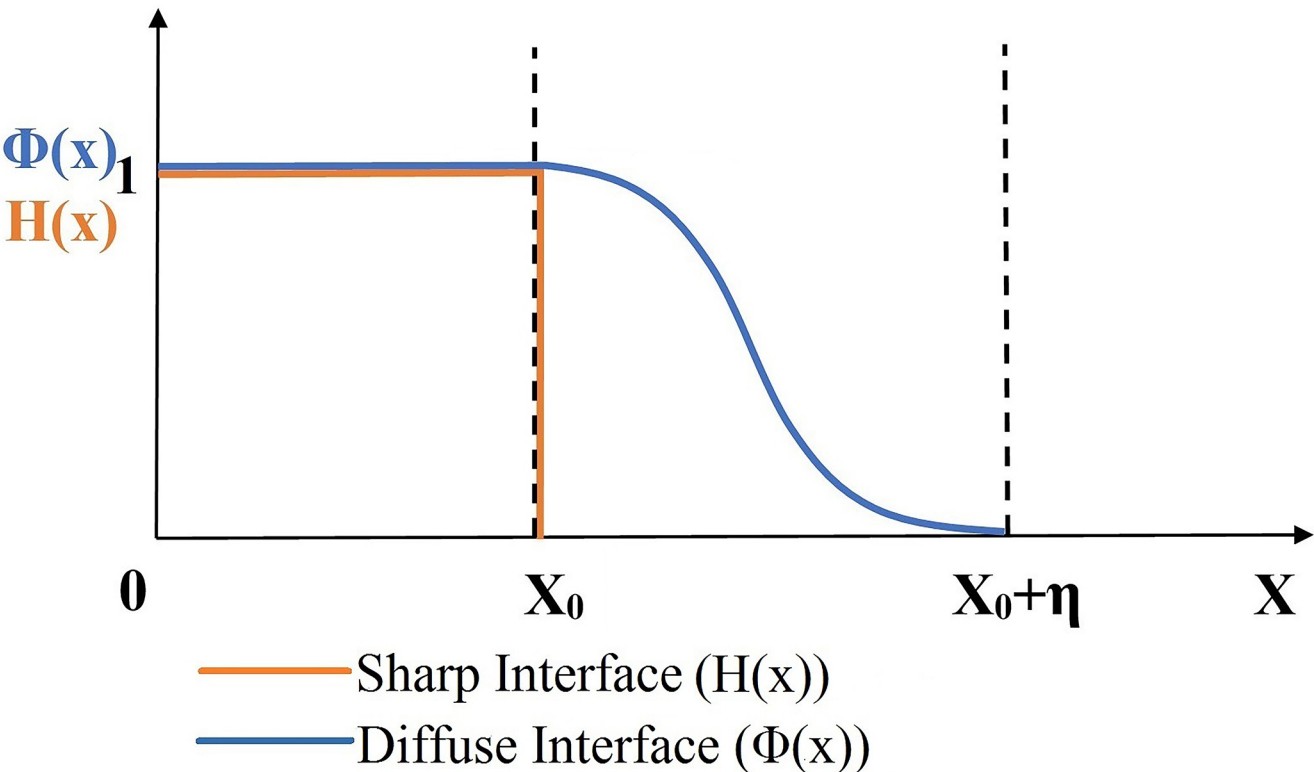

**Fig 1. Visualization of the Heaviside step function H(x) (in orange line) and the phase-field functions Φ(x) (in blue line).**

Thus, the area $A$ of a circle with a radius $r_0$ in circular coordinates is given by $H(x)$ as:

$$A = \iint H(r - r_0) r dr d\theta \qquad (1)$$

The sharp interface property of $H(x)$ reduces the limits of the integral from zero to $r_0$:

$$A = 2\pi \int_0^\infty H(r - r_0) r dr = 2\pi \int_0^{r_0} r dr = \pi r_0^2 \qquad (2)$$

Similarly, the perimeter $P$ of the circle can be calculated using the gradient of $H(x)$. Since the distributional derivative of the $H(x)$ results in the definition of the Dirac delta function ($\delta(x)$) [29], gradients of $H(x)$ will only appear at the boundaries ($r_0$) of the circular object. i.e

$$\delta(x) = \frac{dH(x)}{dx} \qquad (3)$$

The perimeter $P$ of the circle therefore calculated as:

$$P = \iint \delta(r - r_0) r dr d\theta = 2\pi r_0 \qquad (4)$$

In contrast to the $H(x)$, Phase-field functions ($\Phi(x)$) are based on a continuous transition between two states, '1' (presence of the object) and '0' (absence of object), within a small transition zone $\eta$ (Fig 1). However, $\Phi(x)$ can be treated as a continuous and 2-D formulation of $H(x)$ at a very narrow transition width $\eta$.

$$\Phi(r - r_0) \approx H(r - r_0) \qquad (5)$$

and

$$\nabla\Phi(r - r_0) \approx \nabla H(r - r_0) = \delta(r - r_0) \qquad (6)$$

The gradient $\nabla$ describes the one-dimensional derivative in the radial direction.

The following section derives the geometric entropy of circular objects based on the geometric description of the transition region (diffuse interface) in a phase-field function.

## Geometric entropy of circular object

Entropy is a ubiquitous concept that reveals a complicated picture in almost all fields ranging from information [30, 31], thermodynamics [32, 33], biology [34, 35], materials [36, 37], and economics [38, 39]. While some of these are probability distribution functions, others are not. However, irrespective of the notion, all are defined in terms of the well-known logarithmic expression:

$$S = -\sum_{i=0}^N \Phi_i ln \Phi_i \qquad (7)$$

Here we consider the boundary of a circular object as a diffuse interface, which assumes an ideal mixing of an infinite number of equiprobable states of object/non-object ($\Phi_i$). The geometric entropy ($S_{GE}$) at the circle's interface (phase field) is deduced by the geometric description of the diffuse interface using the Temkin model [40].

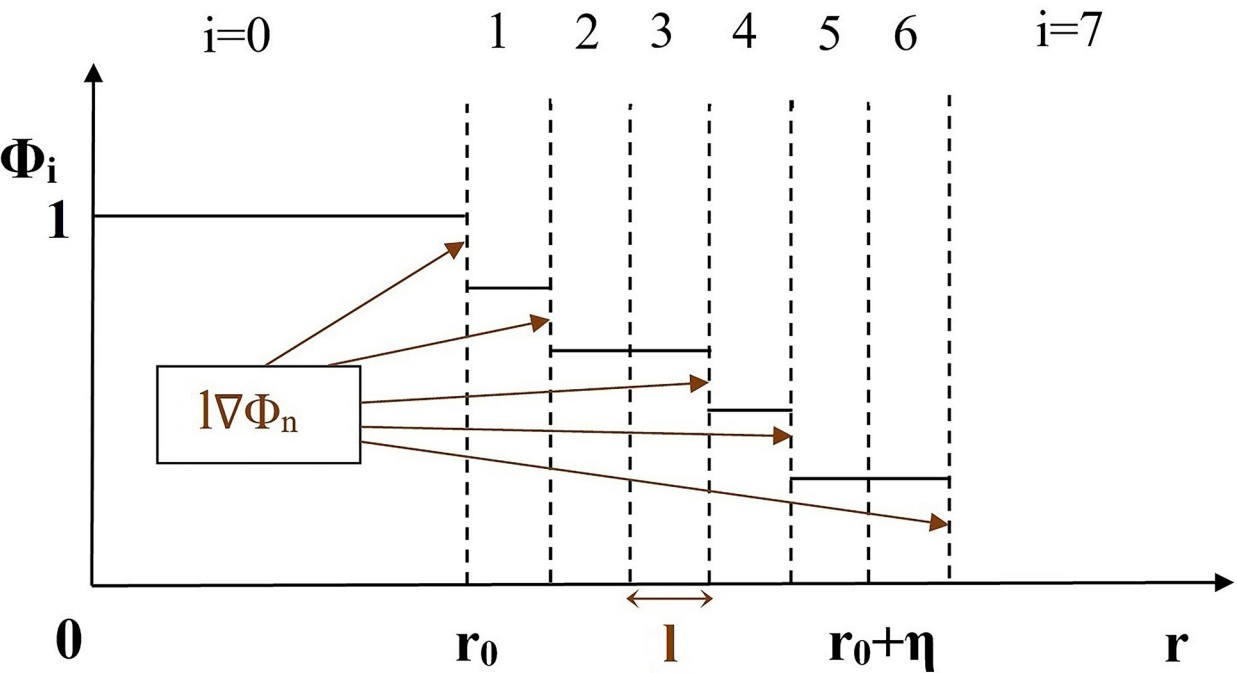

**Fig 2. Visualization of the gradient $\Phi_{n-1} - \Phi_n$ in Temkin's model of the entropy of a diffuse interface (Image concept from Schmitz [22]).**

The Temkin model describes the entropy of the diffuse interface layers as:

$$S = -\sum_{n=-\infty}^{+\infty}(\Phi_{n-1} - \Phi_n)ln(\Phi_{n-1} - \Phi_n) \tag{8}$$

The corresponding gradient of the probable states ($\Phi_i$) can be defined by introducing the notion of discretization length ($l$) between two adjacent layers as (Fig 2):

$$d\Phi_n = \Phi_n - \Phi_{n-1} = \int_{(n-1)l}^{nl}\frac{d\Phi}{dr}\,dr = \frac{d\Phi_n}{dr}\int_{(n-1)l}^{nl}dr = l\frac{d\Phi_n}{dr} = l\nabla_r^n\Phi \tag{9}$$

Therefore, the discrete entropy described in Eq (8) can be transformed into the continuous formulation for small discretization length ($l$) and the infinite number of discrete cells ($n$) as:

$$S = -\sum_{n=-\infty}^{+\infty}\left\{l\frac{d\Phi(nl)}{dr}\right\}ln\left\{l\frac{d\Phi(nl)}{dr}\right\} \Rightarrow \int_{-\infty}^{+\infty}\left\{l\frac{d\Phi(nl)}{dr}\right\}ln\left\{l\frac{d\Phi(nl)}{dr}\right\}dn \tag{10}$$

$$\text{with} \qquad r(n) = r_0 + nl \qquad \text{and} \qquad dn = \frac{dr}{l} \tag{11}$$

Applying Eq (9)

$$S = -\int_{-\infty}^{+\infty}\{l\nabla_r\Phi(r - r_0)\}ln\{l\nabla_r\Phi(r - r_0)\}\frac{dr}{l} \tag{12}$$

Extending the entropy from one dimension to two dimensions in the Cartesian plane changes the radial product $l\nabla_r$ into the scalar product $\vec{l}\,\vec{\nabla}\phi$ and normalizes the integration

direction by some discretization length as:

$$S = -\iint\limits_{-\infty}^{+\infty} (\overrightarrow{l}\overrightarrow{\nabla}\phi)ln\left(\overrightarrow{l}\overrightarrow{\nabla}\phi\right)\frac{dx}{l_x}\frac{dy}{l_y} \tag{13}$$

We recall that the entropy formulation is purely geometrical and does not reveal any intrinsic structure (attributes) of the physical object. Thus, the discretization lengths in the two dimensions can be considered to be a generalized isotropic discretization.

Taking isotropy of discretization, i.e., $l_x = l_y = l_p$, changes the expression of $S$ in the Cartesian plane into polar coordinates by:

$$S = -\iint (\overrightarrow{l}\overrightarrow{\nabla}\phi)ln(\overrightarrow{l}\overrightarrow{\nabla}\phi)\frac{rdrd\theta}{l_p^2} \tag{14}$$

Integrating over the angle $d\theta$ will give:

$$S = -2\pi \int_0^\infty (\overrightarrow{l}\overrightarrow{\nabla}\phi(r - r_0))ln(\overrightarrow{l}\overrightarrow{\nabla}\phi(r - r_0))\frac{rdr}{l_p^2} \tag{15}$$

Since finite values of $\nabla\phi$ can contribute to the integral only at the interface, a proportionality between the terms containing $\nabla\phi$ and the $\delta$-function can be assumed for $\phi$ at small transition widths $\eta$.

$$\frac{l}{l_p}(\overrightarrow{l}\overrightarrow{\nabla}\phi(r - r_0))ln(\overrightarrow{l}\overrightarrow{\nabla}\phi(r - r_0)) \propto \delta(r - r_0) \tag{16}$$

Including Eq (16) with an unknown constant of proportionality into Eq (15) yields:

$$S = -\frac{2\pi}{l_p}\int_0^\infty const \times \delta(r - r_0)rdr = -const \times \frac{2\pi r_0}{l_p} = -const \times \frac{P}{l_p} \tag{17}$$

where $P$ is the perimeter of the circle.

The proportionality constant is estimated as -1/4 in the formulation of the Bekenstein-Hawking entropy of black holes by calculating the average gradient in diffuse interfaces [22].

Hence, the geometric entropy of a circle ($S_{GE}$) takes the dimensionless form:

$$S_{GE} = \frac{1}{4} \times \frac{P}{l_p} \tag{18}$$

## Geometric entropy of plant leaf

Bulk and boundary constitute the structure of any 2-D object. $S_{GE}$ developed at the 2-D interface is purely based on an informational approach and does not contain any temperature term. It is a configurational entropy that conveys information about shape (geometrical features). This information conveys the extent of complexity/dissimilarity of shapes from the circularity.

Geometric entropy is not limited to circular objects. It can be constructed for objects of any shape dimension [22]. In 2-D, an arc is part of the circumference of a circle. Since the area element in plane-polar coordinates is simple ($rdrd\theta$), $S_{GE}$ can be determined directly from Temkin's model. Apart from a circle, every 2-D shape can be visualized as made of infinitesimally small arcs. Hence, their geometric entropy can be arrived at from Temkin's model. However, the progression of the area element ($rdrd\theta$) in plane-polar coordinates complicates the formulation of the entropy of non-circular objects. Notwithstanding the above, the general structure of the geometric entropy of any 2-D shape will be similar to that of a circle ($S_{GE}$) (Eq 18). The

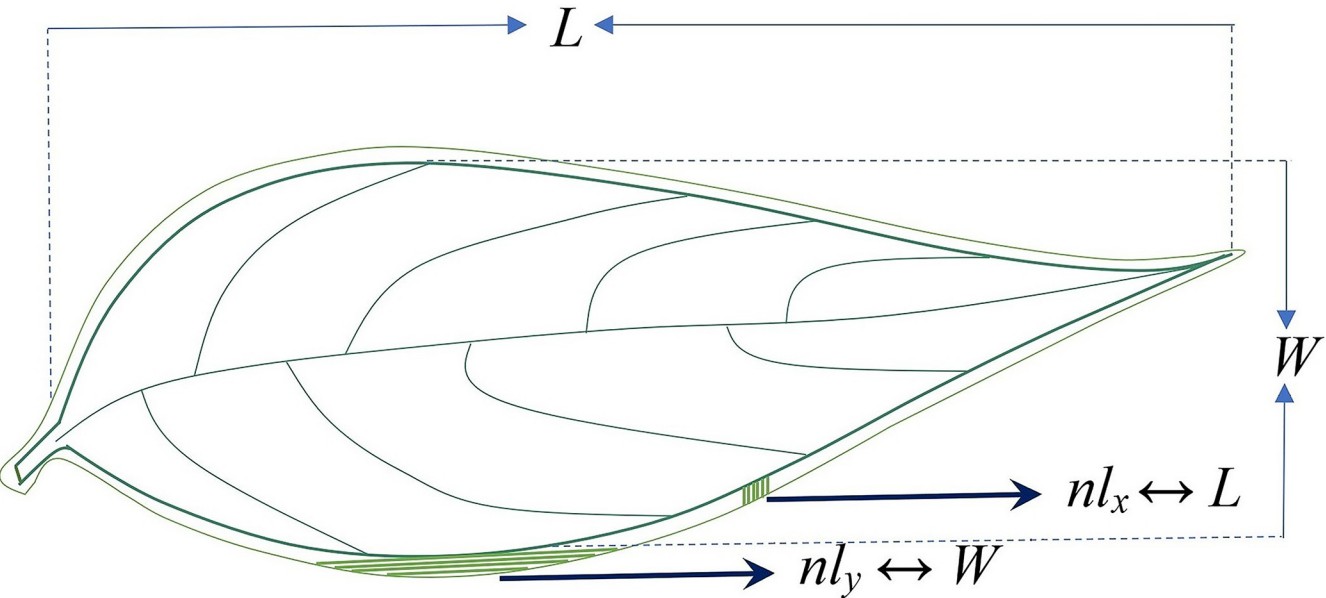

**Fig 3. Visualization of a narrow leaf-environment diffuse interface.** We assume the dimensions of the diffuse interface (in $x$ or $y$ direction) correspond to the leaf dimensions (length $L$ or width $W$).

parameters (circumference and discretization length) will remain the same, but the coefficient in the $S_{GE}$ equation is subject to change. Hence the entropy of any 2-D object is comparable to $S_{GE}$. Here we focus on terrestrial plant leaves viewed as 2-D objects. We consider the leaf-environment interface of the plant leaf as a narrow phase field (Fig 3) and describe the structure of the geometric entropy of a plant leaf ($S_L$) by the $S_{GE}$ of a circular object.

Generally, 2-D shape features are described only in the $x$ and $y$ direction. The parameters of $S_L$ are subjective to the rationale of 2-D complexity and leaf characteristics. We consider the dimensions of the diffuse interface (in $x$ or $y$ direction) to correspond to the leaf dimensions (length $L$ or width $W$) (Fig 3). Finite element method (*FEM*) is an innovative concept to model the development of phase field surfaces. Taking the logic of isotropy of discretization in 2-D non-Euclidean shapes from the finite element method [41], the generalized discretization length ($l_p$) corresponds with the square root of the leaf area. Since the perimeter and leaf area play an important role in leaf physiology [42, 43], we consider the two variables in geometric entropy ($S_L$), perimeter and discretization length; as leaf traits. The perimeter is considered the leaf perimeter, and the discretization length is the square root of the leaf area.

Hence, the geometric entropy ($S_L$) of an ordinary leaf takes the form:

$$S_L = \frac{1}{4} \times \frac{P}{\sqrt{A}} \tag{19}$$

where $P$ is the leaf perimeter, and $A$ is the leaf area.

Eq (19) resembles the leaf dissection index (*LDI*) [44]. While *LDI* is commonly used to depict the complexity of plant leaf shape, we could not trace the assumptions and scientific derivation of *LDI*. Hence we consider Eq (19) as arriving at the geometric entropy through a scientific approach. In the following sections, we illustrate the use of geometric entropy ($S_L$) to analyze the complexity of plant leaf shapes.

## Dataset and measurement

Mature, healthy leaves of 112 flat-leaved plant species in 40 families were collected from Trivandrum (8˚ 32' N, 77˚ 16' E and 8˚ 46' N, 76˚ 41' E), Kerala, India, from June to December 2022 (Table 1). Trivandrum is situated on the southwest coast of India and has a tropical climate with diverse flora and fauna.

Leaves with petioles were scanned on a white background using a digital scanner (Epson L 360). The original RGB images were scaled to $1024 \times 1024$ pixels and converted into bitmap format (24-bit). The scaled color images were transformed into grayscale images and converted to binary images using Otsu's threshold [45]. The binary images were used to estimate the geometric entropy.

The perimeter and area of the binary leaf images were computed using MATLAB [46]. The area of the leaf corresponds to the total number of pixels in the leaf part in the binary images. However, the perimeter corresponds to the number of pixels along the periphery of the leaf parts. The geometric entropy ($S_L$) of the leaves was calculated using Eq (19).

Segmental fractal complexity ($D_{\Sigma S}$) is an improved leaf complexity measure from a fractal-thermodynamic system analogy [47]. It is an algebraic combination of the fractal dimensions of the components of the leaf images, viz., leaf lamina, the background, and leaf edges.

$$D_{\Sigma S} = D_{Background} + D_{Edge} - D_{Leaf} \tag{20}$$

where $D_{Background}$, $D_{Edge,}$ and $D_{Leaf}$ are the fractal dimensions of leaf background, edges, and lamina, respectively. $D_{\Sigma S}$ of the leaves was calculated using Eq (20). $S_L$ was correlated with $D_{\Sigma S}$.

## Results

Plant leaves used in this study exhibited remarkable morphological complexity (Fig 4). The geometric entropy of the leaf images ($S_L$) varied between 2.581 and 30.683 (Table 1). The leaf of *Bridelia retusa* showed the lowest $S_L$, and that of *Jacaranda mimosifolia*, the highest. Deeply lobed broad leaves of *Bauhinia purpurea* and *Merremia vitifolia* showed lower $S_L$ than many simple leaves in the study. Narrow leaves of *Plumeria rubra*, *Acacia auriculiformis*, *Mangifera indica*, *Monoon longifolium*, *Syzygium jambos*, and *Nerium oleander* showed higher $S_L$ than deeply lobed leaves of *Tithonia diversifolia*, *Rhaphidophora tetrasperma*. The $S_L$ of pinnately compound leaves of *Cassia fistula*, *Phyllanthus acidus*, *Senna occidentalis*, *Calliandra haematocephala*, *Simarouba glauca*, *Senna siamea*, *Sesbania grandiflora*, *Azadirachta indica*, *Murraya koenigii*, *Averrhoa bilimbi*, *Caesalpinia pulcherrima*, *Tagetes erecta*, *Caesalpinia coriaria*, *Caesalpinia sappan*, *Albizia odoratissima*, *Melia azedarach*, *Moringa oleifera*, *Phyllanthus emblica*, and *Jacaranda mimosifolia* were the highest and ranged from 6.158 to 30.683 (Table 1). The $S_L$ values increase with decreasing leaf width and increasing leaf pinnation. $S_L$ was comparable for leaves with similar morphology.

Table 1 presents the segmental fractal complexity ($D_{\Sigma S}$) of the leaves studied. $D_{\Sigma S}$ varies between 1.044 and 1.952. The pinnately compound leaf of *Jacaranda mimosifolia* recorded the highest $D_{\Sigma S}$. The lowest $D_{\Sigma S}$ was observed for the simple leaf of *Calotropis gigantea*. The $D_{\Sigma S}$ of the pinnately compound leaves of *Senna occidentalis*, *Simarouba glauca*, *Albizia odoratissima*, *Sesbania grandiflora*, *Azadirachta indica*, *Calliandra haematocephala*, *Cassia fistula*, *Murraya koenigii*, *Melia azedarach*, *Moringa oleifera*, *Averrhoa bilimbi*, *Tagetes erecta*, *Phyllanthus emblica*, and *Jacaranda mimosifolia* ranged between 1.215 and 1.952 (Table 1). $D_{\Sigma S}$ also increases with decreasing leaf width and increasing leaf lobiness and pinnation. Fig 5 illustrates a strong monotonic relationship ($\rho = 0.95$, p < 0.001) between $S_L$ and $D_{\Sigma S}$.

We classified the variation in $S_L$ at the taxonomic level. Fig 6 depicts the similarity dendrogram. The high cophenetic correlation coefficient (0.97) indicates the quality of classification.

**Table 1. Geometric entropy (S$_L$) and segmental fractal complexity (D$_{\Sigma S}$) of the plant leaves collected from Trivandrum, Kerala, India.**

| Sl. No. | Plant species | Genus | Family | Segmental fractal complexity ($D_{\Sigma S}$) | Geometric entropy ($S_L$) | Leaf type |
|---|---|---|---|---|---|---|
| 1 | *Justicia adhatoda L.* | Justicia | Acanthaceae | 1.105 | 3.876 | Simple |
| 2 | *Hydnocarpus pentandrus (Buch.-Ham.) Oken* | Hydnocarpus | Achariaceae | 1.099 | 3.807 | Simple |
| 3 | *Anacardium occidentale L.* | Anacardium | Anacardiaceae | 1.057 | 2.73 | Simple |
| 4 | *Mangifera indica L.* | Mangifera | Anacardiaceae | 1.12 | 4.262 | Simple, Narrow |
| 5 | *Spondias pinnata (L.fil.) Kurz* | Spondias | Anacardiaceae | 1.092 | 3.215 | Simple |
| 6 | *Annona muricata L.* | Annona | Annonaceae | 1.067 | 3.029 | Simple |
| 7 | *Annona reticulata L.* | Annona | Annonaceae | 1.087 | 3.417 | Simple |
| 8 | *Annona squamosa L.* | Annona | Annonaceae | 1.081 | 3.413 | Simple |
| 9 | *Monoon longifolium (Sonn.) B.Xue & R.M. K.Saunders* | Monoon | Annonaceae | 1.121 | 4.312 | Simple, Narrow |
| 10 | *Alstonia scholaris (L.) R. Br.* | Alstonia | Apocynaceae | 1.099 | 3.701 | Simple |
| 11 | *Calotropis gigantea (L.) W.T.Aiton* | Calotropis | Apocynaceae | 1.044 | 2.636 | Simple |
| 12 | *Plumeria rubra L.* | Plumeria | Apocynaceae | 1.114 | 4.000 | Simple, Narrow |
| 13 | *Tabernaemontana alternifolia L.* | Tabernaemontana | Apocynaceae | 1.103 | 3.821 | Simple |
| 14 | *Nerium oleander L.* | Nerium | Apocynaceae | 1.119 | 4.652 | Simple, Narrow |
| 15 | *Plumeria alba L.* | Plumeria | Apocynaceae | 1.082 | 3.162 | Simple |
| 16 | *Rhaphidophora tetrasperma Hook.f.* | Rhaphidophora | Araceae | 1.081 | 3.972 | Simple, Lobed |
| 17 | *Tagetes erecta L.* | Tagetes | Asteraceae | 1.408 | 8.911 | Unipinnate |
| 18 | *Tithonia diversifolia (Hemsl.) A.Gray* | Tithonia | Asteraceae | 1.108 | 3.903 | Simple, Lobed |
| 19 | *Handroanthus impetiginosus (Mart. ex DC.) Mattos* | Handroanthus | Bignoniaceae | 1.100 | 3.594 | Simple |
| 20 | *Jacaranda mimosifolia D.Don* | Jacaranda | Bignoniaceae | 1.952 | 30.683 | Bipinnate |
| 21 | *Tecoma stans (L.) Juss. ex Kunth* | Tecoma | Bignoniaceae | 1.114 | 3.685 | Simple, Toothed |
| 22 | *Trema tomentosa (Roxb.) H. Hara* | Trema | Cannabaceae | 1.122 | 4.171 | Simple, Toothed |
| 23 | *Carica papaya L.* | Carica | Caricaceae | 1.135 | 4.44 | Simple, Lobed |
| 24 | *Garcinia mangostana L.* | Garcinia | Clusiaceae | 1.103 | 3.578 | Simple |
| 25 | *Terminalia arjuna (Roxb. ex DC.) Wight & Arn.* | Terminalia | Combretaceae | 1.082 | 3.619 | Simple |
| 26 | *Terminalia bellirica (Gaertn.) Roxb.* | Terminalia | Combretaceae | 1.068 | 3.044 | Simple |
| 27 | *Merremia vitifolia (Burm. f.) Hallier f.* | Merremia | Convolvulaceae | 1.077 | 3.177 | Simple, Lobed |
| 28 | *Hopea ponga (Dennst.) Mabb.* | Hopea | Dipterocarpaceae | 1.098 | 3.452 | Simple |
| 29 | *Hopea parviflora Bedd.* | Hopea | Dipterocarpaceae | 1.080 | 3.281 | Simple |
| 30 | *Vateria indica L.* | Vateria | Dipterocarpaceae | 1.120 | 3.437 | Simple, Narrow |
| 31 | *Hevea brasiliensis (Willd. ex A.Juss.) Müll. Arg.* | Hevea | Euphorbiaceae | 1.058 | 3.242 | Simple |
| 32 | *Macaranga peltata (Roxb.) Müll.Arg.* | Macaranga | Euphorbiaceae | 1.071 | 3.066 | Simple |
| 33 | *Acacia auriculiformis Benth.* | Acacia | Fabaceae | 1.127 | 4.208 | Simple, Narrow |
| 34 | *Acacia mangium Willd.* | Acacia | Fabaceae | 1.089 | 3.717 | Simple |
| 35 | *Albizia odoratissima (L.f.) Benth.* | Albizia | Fabaceae | 1.226 | 10.14 | Bipinnate |
| 36 | *Bauhinia purpurea L.* | Bauhinia | Fabaceae | 1.048 | 2.719 | Simple, Lobed |
| 37 | *Caesalpinia sappan L.* | Caesalpinia | Fabaceae | 1.173 | 9.299 | Bipinnate |
| 38 | *Caesalpinia pulcherrima (L.) Sw.* | Caesalpinia | Fabaceae | 1.16 | 8.54 | Bipinnate |
| 39 | *Caesalpinia coriaria (Jacq.) Willd.* | Caesalpinia | Fabaceae | 1.159 | 9.040 | Bipinnate |
| 40 | *Calliandra haematocephala Hassk.* | Calliandra | Fabaceae | 1.233 | 6.356 | Unipinnate |
| 41 | *Cassia fistula L.* | Cassia | Fabaceae | 1.233 | 6.158 | Unipinnate |
| 42 | *Pueraria phaseoloides (Roxb.) Benth.* | Pueraria | Fabaceae | 1.116 | 4.004 | Trifoliate |
| 43 | *Gliricidia sepium (Jacq.) Kunth* | Gliricidia | Fabaceae | 1.15 | 5.863 | Unipinnate |

*(Continued)*

**Table 1.** (Continued)

| Sl. No. | Plant species | Genus | Family | Segmental fractal complexity ($D_{\Sigma S}$) | Geometric entropy ($S_L$) | Leaf type |
|---|---|---|---|---|---|---|
| 44 | *Indigofera hirsuta L.* | Indigofera | Fabaceae | 1.206 | 5.574 | Unipinnate |
| 45 | *Senna occidentalis L.* | Senna | Fabaceae | 1.215 | 6.333 | Unipinnate |
| 46 | *Senna siamea (Lam.) H.S.Irwin & Barneby* | Senna | Fabaceae | 1.186 | 6.896 | Unipinnate |
| 47 | *Senna surattensis (Burm.f.) H.S.Irwin & Barneby* | Senna | Fabaceae | 1.197 | 5.192 | Unipinnate |
| 48 | *Sesbania grandiflora (L.) Pers.* | Sesbania | Fabaceae | 1.227 | 6.979 | Unipinnate |
| 49 | *Tamarindus indica L.* | Tamarindus | Fabaceae | 1.125 | 5.328 | Unipinnate |
| 50 | *Holmskioldia sanguinea Retz.* | Holmskioldia | Lamiaceae | 1.077 | 3.218 | Simple, Toothed |
| 51 | *Vitex negundo L.* | Vitex | Lamiaceae | 1.211 | 6.034 | Palmate, 3–5 foliolate |
| 52 | *Cinnamomum tamala (Buch.-Ham.) T. Nees & Eberm.* | Cinnamomum | Lauraceae | 1.098 | 3.515 | Simple |
| 53 | *Cinnamomum verum J.Presl* | Cinnamomum | Lauraceae | 1.055 | 2.852 | Simple |
| 54 | *Strychnos nux-vomica L.* | Strychnos | Loganiaceae | 1.08 | 3.1 | Simple |
| 55 | *Dendrophthoe falcata (L.fil.) Blume* | Dendrophthoe | Loranthaceae | 1.092 | 3.874 | Simple |
| 56 | *Lagerstroemia speciosa (L.) Pers.* | Lagerstroemia | Lythraceae | 1.051 | 2.734 | Simple |
| 57 | *Lawsonia inermis L.* | Lawsonia | Lythraceae | 1.068 | 2.890 | Simple |
| 58 | *Durio zibethinus Murray* | Durio | Malvaceae | 1.102 | 3.528 | Simple |
| 59 | *Hibiscus cannabinus L.* | Hibiscus | Malvaceae | 1.204 | 5.591 | Simple, Lobed |
| 60 | *Hibiscus rosa-sinensis L.* | Hibiscus | Malvaceae | 1.112 | 3.825 | Simple, Toothed |
| 61 | *Hibiscus tiliaceus L.* | Hibiscus | Malvaceae | 1.082 | 2.968 | Simple |
| 62 | *Sterculia balanghas L.* | Sterculia | Malvaceae | 1.079 | 3.148 | Simple |
| 63 | *Thespesia populnea (L.) Sol. ex Corrêa* | Thespesia | Malvaceae | 1.102 | 3.408 | Simple |
| 64 | *Clidemia hirta (L.) D. Don* | Clidemia | Melastomataceae | 1.059 | 2.716 | Simple, Toothed |
| 65 | *Azadirachta indica A.Juss.* | Azadirachta | Meliaceae | 1.23 | 7.642 | Unipinnate |
| 66 | *Melia azedarach L.* | Melia | Meliaceae | 1.352 | 10.292 | Bipinnate |
| 67 | *Artocarpus heterophyllus Lam.* | Artocarpus | Moraceae | 1.076 | 2.964 | Simple |
| 68 | *Artocarpus hirsutus Lam.* | Artocarpus | Moraceae | 1.049 | 2.686 | Simple |
| 69 | *Ficus benghalensis L.* | Ficus | Moraceae | 1.053 | 2.652 | Simple |
| 70 | *Ficus elastica Roxb.* | Ficus | Moraceae | 1.057 | 2.779 | Simple |
| 71 | *Ficus exasperata Vahl* | Ficus | Moraceae | 1.088 | 3.41 | Simple |
| 72 | *Ficus hispida L.fil.* | Ficus | Moraceae | 1.080 | 3.201 | Simple |
| 73 | *Ficus religiosa L.* | Ficus | Moraceae | 1.125 | 3.842 | Simple |
| 74 | *Morus alba L.* | Morus) | Moraceae | 1.051 | 2.722 | Simple, Toothed |
| 75 | *Morus macroura Miq.* | Morus | Moraceae | 1.069 | 3.113 | Simple, Toothed |
| 76 | *Moringa oleifera Lam.* | Moringa | Moringaceae | 1.384 | 10.954 | Tripinnate |
| 77 | *Eugenia victoriana Cuatrec.* | Eugenia | Myrtaceae | 1.100 | 3.615 | Simple |
| 78 | *Pimenta dioica (L.) Merr.* | Pimenta | Myrtaceae | 1.066 | 2.854 | Simple |
| 79 | *Psidium cattleianum Afzel. ex Sabine* | Psidium | Myrtaceae | 1.060 | 2.987 | Simple |
| 80 | *Psidium guajava L.* | Psidium | Myrtaceae | 1.074 | 3.131 | Simple |
| 81 | *Syzygium aqueum (Burm.fil.) Alston* | Syzygium | Myrtaceae | 1.063 | 2.716 | Simple |
| 82 | *Syzygium aromaticum (L.) Merr. & Perry* | Syzygium | Myrtaceae | 1.090 | 3.545 | Simple |
| 83 | *Syzygium cumini (L.) Skeels* | Syzygium | Myrtaceae | 1.078 | 3.045 | Simple |
| 84 | *Syzygium jambos (L.) Alston* | Syzygium | Myrtaceae | 1.144 | 4.522 | Simple, Narrow |
| 85 | *Syzygium malaccense (L.) Merr. & L.M. Perry* | Syzygium | Myrtaceae | 1.092 | 3.511 | Simple |

(*Continued*)

**Table 1.** (Continued)

| Sl. No. | Plant species | Genus | Family | Segmental fractal complexity ($D_{\Sigma S}$) | Geometric entropy ($S_L$) | Leaf type |
|---|---|---|---|---|---|---|
| 86 | *Syzygium samarangense* (Blume) Merr. & L.M.Perry | Syzygium | Myrtaceae | 1.071 | 2.992 | Simple |
| 87 | *Nyctanthes arbor-tristis* L. | Nyctanthes | Oleaceae | 1.057 | 3.064 | Simple |
| 88 | *Averrhoa bilimbi* L. | Averrhoa | Oxalidaceae | 1.403 | 8.395 | Unipinnate |
| 89 | *Averrhoa carambola* L. | Averrhoa | Oxalidaceae | 1.117 | 4.812 | Unipinnate |
| 90 | *Bridelia retusa* (L.) A.Juss. | Bridelia | Phyllanthaceae | 1.051 | 2.581 | Simple |
| 91 | *Phyllanthus acidus* (L.) Skeels | Phyllanthus | Phyllanthaceae | 1.19 | 6.188 | Unipinnate |
| 92 | *Phyllanthus emblica* L. | Phyllanthus | Phyllanthaceae | 1.579 | 13.862 | Unipinnate |
| 93 | *Sauropus androgynus* (L.) Merr. | Sauropus | Phyllanthaceae | 1.135 | 5.832 | Unipinnate |
| 94 | *Piper longum* L. | Piper | Piperaceae | 1.076 | 3.279 | Simple |
| 95 | *Piper nigrum* L. | Piper | Piperaceae | 1.064 | 2.833 | Simple |
| 96 | *Xanthophyllum flavescens* Roxb. | Xanthophyllum | Polygalaceae | 1.085 | 3.295 | Simple |
| 97 | *Carallia brachiata* (Lour.) Merr. | Carallia | Rhizophoraceae | 1.062 | 3.174 | Simple |
| 98 | *Mussaenda philippica* A.Rich. | Mussaenda | Rubiaceae | 1.087 | 3.283 | Simple |
| 99 | *Aegle marmelos* (L.) Corrêa | Aegle | Rutaceae | 1.21 | 5.976 | Compound, 3–5 foliolate |
| 100 | *Murraya koenigii* (L.) Spreng. | Murraya | Rutaceae | 1.245 | 7.787 | Unipinnate |
| 101 | *Flacourtia jangomas* (Lour.) Raeusch. | Flacourtia | Salicaceae | 1.056 | 2.786 | Simple, Toothed |
| 102 | *Flacourtia sepiaria* Roxb. | Flacourtia | Salicaceae | 1.078 | 3.031 | Simple, Toothed |
| 103 | *Santalum album* L. | Santalum | Santalaceae | 1.091 | 3.217 | Simple |
| 104 | *Nephelium lappaceum* L. | Nephelium | Sapindaceae | 1.106 | 2.806 | Simple |
| 105 | *Nephelium mutabile* Blume | Nephelium | Sapindaceae | 1.086 | 3.456 | Simple |
| 106 | *Chrysophyllum cainito* L. | Chrysophyllum | Sapotaceae | 1.075 | 3.105 | Simple |
| 107 | *Chrysophyllum oliviforme* L. | Chrysophyllum | Sapotaceae | 1.057 | 2.726 | Simple |
| 108 | *Manilkara zapota* (L.) P.Royen | Manilkara | Sapotaceae | 1.122 | 3.882 | Simple |
| 109 | *Mimusops elengi* L. | Mimusops | Sapotaceae | 1.101 | 3.521 | Simple |
| 110 | *Pouteria caimito* (Ruiz & Pav.) Radlk. | Pouteria | Sapotaceae | 1.080 | 3.426 | Simple |
| 111 | *Synsepalum dulcificum* (Schumach. & Thonn.) Daniell | Synsepalum | Sapotaceae | 1.102 | 3.624 | Simple |
| 112 | *Simarouba glauca* DC. | Simarouba | Simaroubaceae | 1.216 | 6.482 | Unipinnate |

The leaves of 112 species were grouped into 5 clusters (indicated with different colors in Fig 6) at 89% similarity threshold. Three-quarters of the species studied were clubbed as cluster 1 (red in Fig 6). Except for *Averrhoa carambola*, all other species in cluster 1 had simple leaves. The similarity of this cluster ranges from 96–100%. $S_L$ in cluster 1 varies between only 2.581 and 4.812. However, the values increase with decreasing leaf width. Narrow simple leaves exhibited comparatively a higher $S_L$ and were clubbed together in cluster 1. Read in an anti-clockwise direction; narrow leaves predominantly occupied the tail end of cluster 1 (Fig 6). The following families included multiple species and were unique to cluster 1: Anacardiaceae (3 species), Annonaceae (4 species), Apocynaceae (6 species), Combretaceae (2 species), Dipterocarpaceae (3 species), Euphorbiaceae (2 species), Lauraceae (2 species), Lythraceae (2 species), Moraceae (9 species), Myrtaceae (10 species), Piperaceae (2 species), Salicaceae (2 species), Sapindaceae (2 species), and Sapotaceae (6 species). Different species from the same genus of the above listed families showed remarkable similarity in shape in this cluster (Table 2).

Cluster 2 (blue in Fig 6) comprises predominantly uni-pinnate species. It does not include any simple or bi or tri-pinnate leaves. The similarity of this cluster ranges between 94–100%.

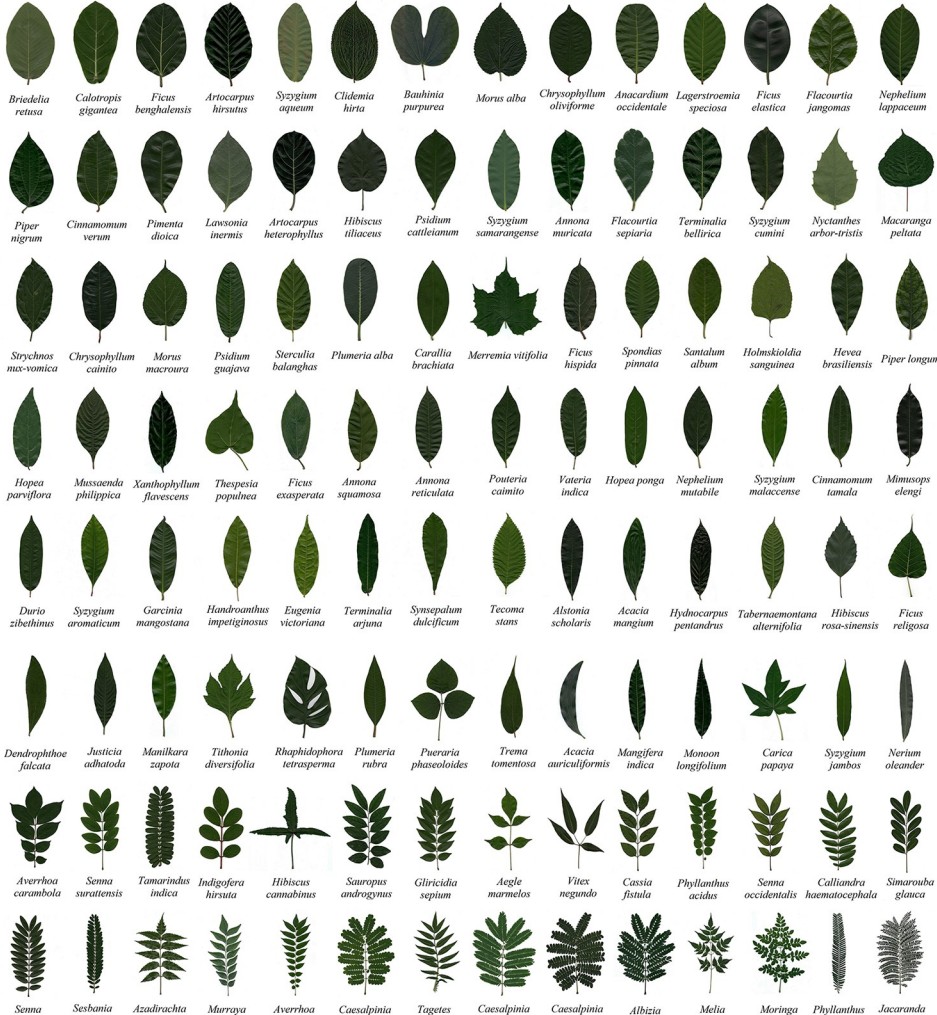

**Fig 4. Morphological complexity of the plant leaves studied.** Arrangements of the leaves are in the increasing order of geometric entropy ($S_L$).

Of these, *Senna surattensis*, *Tamarindus indica*, *Indigofera hirsute*, *Gliricidia sepium*, *Cassia fistula*, *Senna occidentalis*, *Calliandra haematocephala*, *Senna siamea*, and *Sesbania grandiflora* were from Fabaceae family. $S_L$ of leaves of species from the Fabaceae family varies between 2.719 (*Bauhinia purpurea*) and 10.14 (*Albizia odoratissima*). Leaves of species under the Fabaceae family exhibit diverse shapes ranging from simple to bipinnate. The former was categorized into cluster 1 and the latter into cluster 3. Three species from the genus Senna, *Senna surattensis* ($S_L$—5.192), *Senna occidentalis* ($S_L$—6.333), *Senna siamea* ($S_L$—6.896) were grouped in cluster 2 (Table 2).

Cluster 3 (green in Fig 6) comprises only pinnately compound leaves, and their $S_L$ varies between 8.395 and 10.954. The similarity of this cluster ranges from 93% to 100%. Of these *Averrhoa bilimbi*, *Tagetes erecta* were uni-pinnate, and *Caesalpinia pulcherrima*, *Caesalpinia coriaria*, *Caesalpinia sappan*, *Albizia odoratissima*, and *Melia azedarach*, were bi-pinnate. However, the leaf of *Moringa oleifera* ($S_L$—10.954) is tripinnate and is the distant '*leaf*'. Three species from the same genus: Caesalpinia (Fabaceae), *Caesalpinia pulcherrima* ($S_L$—8.540),

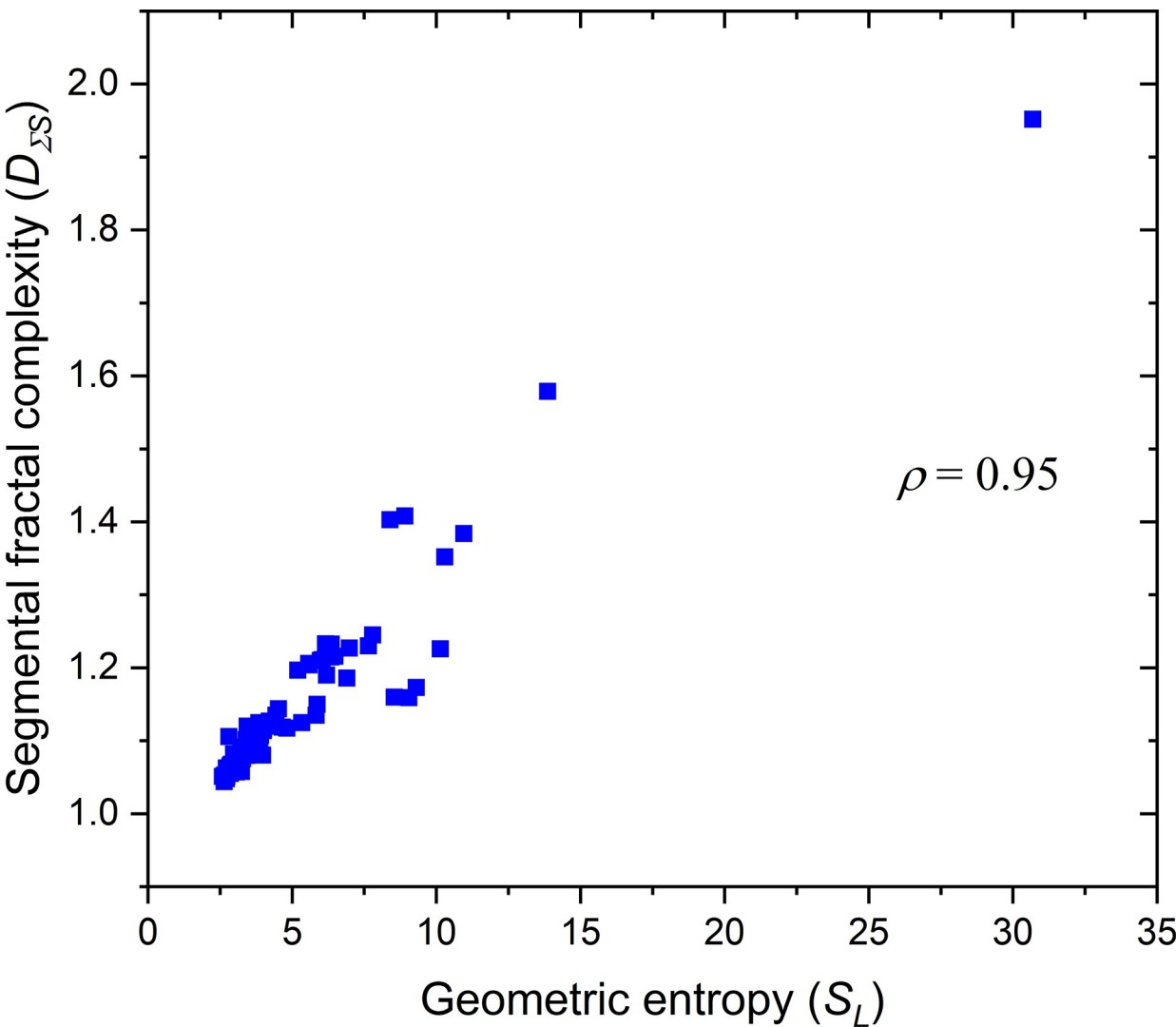

**Fig 5. Monotonic relationship between the geometric entropy ($S_L$) and segmental fractal complexity ($D_{\Sigma S}$).**

*Caesalpinia coriaria* (9.040), and *Caesalpinia sappan* ($S_L$—9.299), were also grouped in cluster 3 (Table 2).

Cluster 4 and 5 (magenta and yellow in Fig 6) constitutes only one clade each, *Phyllanthus emblica* (Phyllanthaceae, $S_L$—13.862), and *Jacaranda mimosifolia* (Bignoniaceae, $S_L$—30.683), respectively, and stand out as *simplicifolious* (dissimilarity 17% and 100% respectively). $S_L$ of species' leaves from Phyllanthaceae varies from 2.581 (*Bridelia retusa*) to 13.862 (*Phyllanthus emblica*). Their shapes range from simple (*Bridelia retusa*) to uni-pinnate (*Sauropus androgynus*, *Phyllanthus acidus*, *Phyllanthus emblica*). Apart from cluster 4, they also belong to cluster 2 (*Sauropus androgynus*, *Phyllanthus acidus*). Similarly, *Handroanthus impetiginosus* ($S_L$—3.594), and *Tecoma stans* ($S_L$—3.685) from Bignoniaceae, were also included in cluster 1. $S_L$ of species from Bignoniaceae varies from 3.594 (*Handroanthus impetiginosus*) to 30.683 (*Jacaranda mimosifolia*). While the present study does not reveal well-differentiated clusters to discriminate $S_L$ at the family level, it does reveal the well-differentiated clustering of $S_L$ at the genus level (Table 2).

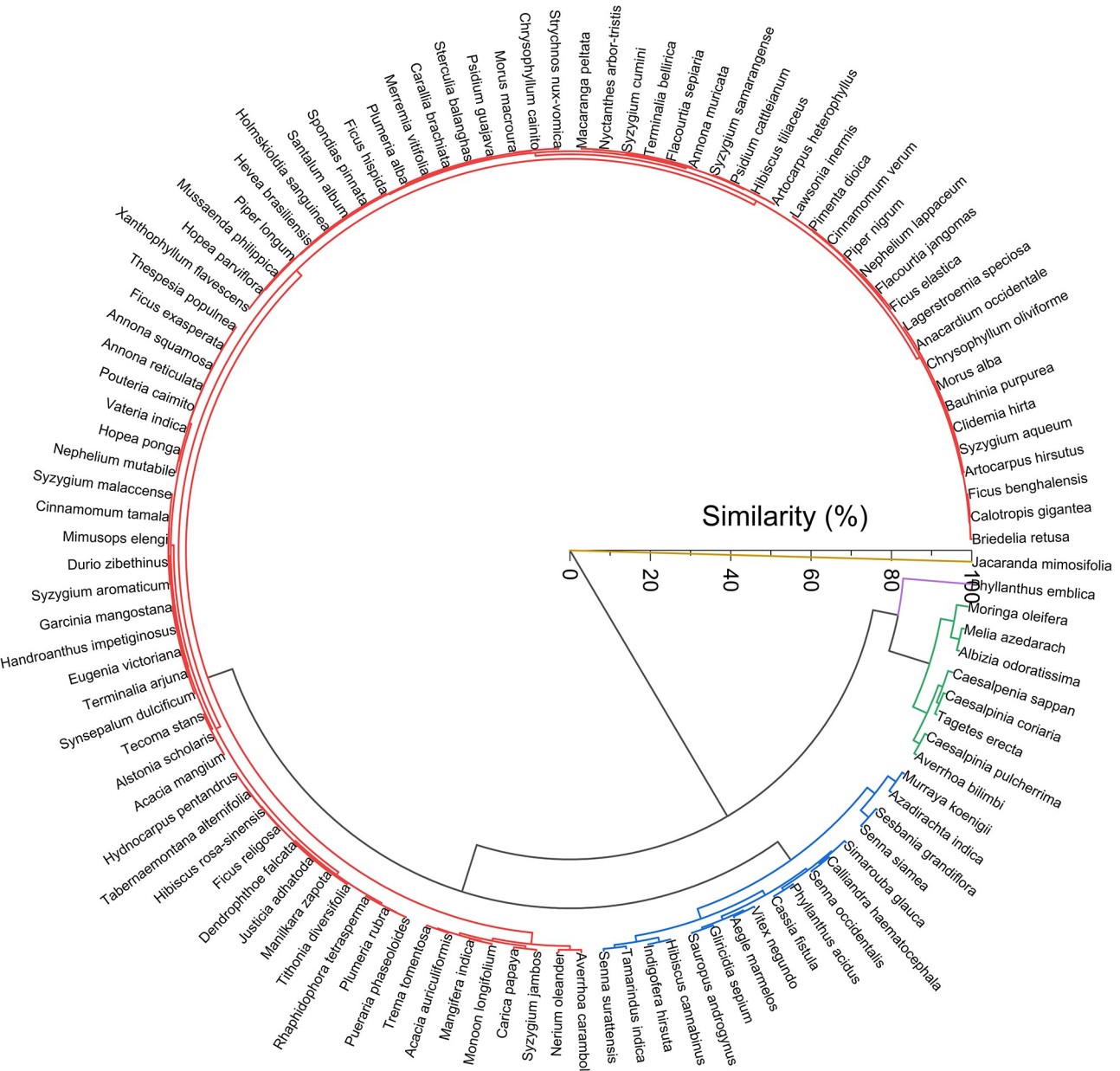

**Fig 6. Dendrogram of geometric entropy of plant leaves ($S_L$).** The dendrogram results in 5 clusters at a cutoff similarity of 89%.

## Discussion

Geometric shape (information) plays a vital role in the ecological system [48]. The morphological structure of life forms carries information imparted by its geometry to distinguish them from others. However, the geometry of every living object is bestowed by the boundary that limits their biomass. Geometric analysis of natural structures allows for determining their shape optimization to maximize energy efficiency and visual appeal [49].

Defining boundaries is vital in morphometric analysis. In current knowledge, living forms are confined only to three dimensions. Boundaries distinguishing organisms (or any objects) from their environment (or non-objects) also have characteristic dimensions. One-

**Table 2. Classification of plant species at genus level based on hierarchical cluster analysis of geometric entropy (S$_L$).**

| Family | Genus | Plant species | | |
|---|---|---|---|---|
| | | **Cluster 1** | **Cluster 2** | **Cluster 3** |
| Annonaceae | Annona | *Annona muricata L.* | | |
| | | *Annona squamosa L.* | | |
| | | *Annona reticulata L.* | | |
| Apocynaceae | Plumeria | *Plumeria rubra L.* | | |
| | | *Plumeria alba L.* | | |
| Combretaceae | Terminalia | *Terminalia bellirica (Gaertn.) Roxb.* | | |
| | | *Terminalia arjuna (Roxb. ex DC.) Wight & Arn.* | | |
| Dipterocarpaceae | Hopea | *Hopea ponga (Dennst.) Mabb.* | | |
| | | *Hopea parviflora Bedd.* | | |
| Fabaceae | Acacia | *Acacia mangium Willd.* | | |
| | | *Acacia auriculiformis Benth.* | | |
| | Senna | | *Senna surattensis (Burm.f.) H.S.Irwin & Barneby* | |
| | | | *Senna occidentalis L.* | |
| | | | *Senna siamea (Lam.) H.S.Irwin & Barneby* | |
| | Caesalpinia | | | *Caesalpinia pulcherrima (L.) Sw.* |
| | | | | *Caesalpinia coriaria (Jacq.) Willd.* |
| | | | | *Caesalpinia sappan L.* |
| Lauraceae | Cinnamomum | *Cinnamomum verum J.Presl* | | |
| | | *Cinnamomum tamala (Buch.-Ham.) T.Nees & Eberm.* | | |
| Malvaceae | Hibiscus | *Hibiscus tiliaceus L.* | | |
| | | *Hibiscus rosa-sinensis L.* | | |
| Moraceae | Ficus | *Ficus benghalensis L.* | | |
| | | *Ficus elastica Roxb.* | | |
| | | *Ficus hispida L.fil.* | | |
| | | *Ficus exasperata Vahl* | | |
| | | *Ficus religiosa L.* | | |
| | Morus | *Morus alba L.* | | |
| | | *Morus macroura Miq.* | | |
| Myrtaceae | Syzygium | *Syzygium aqueum (Burm.fil.) Alston* | | |
| | | *Syzygium samarangense (Blume) Merr. & L.M. Perry* | | |
| | | *Syzygium cumini (L.) Skeels* | | |
| | | *Syzygium malaccense (L.) Merr. & L.M.Perry* | | |
| | | *Syzygium aromaticum (L.) Merr. & Perry* | | |
| | | *Syzygium jambos (L.) Alston* | | |
| | Psidium | *Psidium cattleianum Afzel. ex Sabine* | | |
| | | *Psidium guajava L.* | | |
| Piperaceae | Piper | *Piper nigrum L.* | | |
| | | *Piper longum L.* | | |
| Salicaceae | Flacourtia | *Flacourtia jangomas (Lour.) Raeusch.* | | |
| | | *Flacourtia sepiaria Roxb.* | | |
| Sapindaceae | Nephelium | *Nephelium lappaceum L.* | | |
| | | *Nephelium mutabile Blume* | | |

*(Continued)*

**Table 2.** (Continued)

| Family | Genus | Plant species | | |
|---|---|---|---|---|
| | | Cluster 1 | Cluster 2 | Cluster 3 |
| Sapotaceae | Chrysophyllum | *Chrysophyllum oliviforme L.* | | |
| | | *Chrysophyllum cainito L.* | | |

Clusters 4 and 5 are *simplicifolious* with *Phyllanthus emblica* and *Jacaranda mimosifolia*, respectively. Plant species from the Fabaceae family exhibit diverse leaf shapes across different genera and are thus included in multiple clusters.

dimensional objects are lines confined by a boundary consisting of two points (dimensionless). 2-D objects are planar and confined by a boundary, which is a line (1-D). Similarly, 3-D objects are spatial and confined by a boundary, which is an area (2-D). In general, an n-dimensional object is confined by an n-1 dimensional boundary.

We describe the geometry of a circular object (2-D) by the sharp interface property of the Heaviside step function ($H(x)$) (Fig 1). Considering the boundary of the circular object as a diffuse interface with an infinite number of equiprobable states of presence and absence of the object, the geometric entropy ($S_{GE}$) at the circle's interface is deduced by using a phase-field function (Fig 2). The $S_{GE}$ (in Eq (18)) is composed of only two variables: circumference and discretization length. This entropy is associated only with the information contained in the structural form of objects.

Geometric entropy can be constructed for any object of any dimension. $S_{GE}$ was transformed by considering the plant leaf-environment interface as a narrow phase field to describe the geometric entropy of plant leaves ($S_L$). Two leaf traits, viz., leaf area and perimeter, were selected as the parameters in the relation. They were selected based on their dimensionality and physiological importance.

Plant leaves absorb solar energy and process it to maintain a high organization with lower entropy by photosynthesis [50]. The capture of energy and carbon assimilation in plant leaves depends on the geometry and positioning of the leaf lamina. An increase in leaf perimeter (by serration or lobes) increases the entropy in shape and geometric complexity. However, a high perimeter achieved by increasing the laminar area may not increase the leaf complexity unless accompanied by a significant increase in the number of serrations or lobes. Leaf perimeter influences its physiology in many ways [51–53]. Increasing the marginal serration without compromising the leaf area could be regarded as an adaptive strategy in terms of heating [54]. Deep serrations or lobes reduce the leaf area and maintain photosynthetic tissues closer to veins, thereby supporting high photosynthetic rates [55, 56]. The leaf hydraulic resistance also describes the adaptive stability of dissected leaves. Lobed and dissected leaves have fewer minor veins that reduce hydraulic resistance than entire leaves, which is advantageous in dry environments [55, 56]. Further, leaf margin optimization with air temperature has been confirmed in many areas and utilized in many plant evolutionary studies [57, 58].

Leaf energetics have focused more on leaf area than other traits such as leaf length, width, thickness, and perimeter. A larger leaf area is advantageous in light capture and photosynthetic productivity [59]. The seemingly obvious behavior of small leaves in higher canopies [60] and larger leaves in lower canopies [61, 62] provide evidence of the importance of leaf surface area in solar energy absorption and photosynthesis [63]. Notably, smaller leaves with significant vein density are more efficient in nutrient transport and tolerant to leaf vein embolism [64].

The areal optimization of the leaves also influences the leaf thermal regulation via boundary layer thickness [65, 66]. Larger leaves have a thick leaf boundary layer that diminishes the convective heat loss and gas exchange between the leaf and the surrounding air compared to

smaller leaves with a thin boundary layer [67]. Further, the decrease in leaf size with decreasing water availability [68] reduces the leaf temperature and avoids overheating in arid environments [69]. Therefore, having small leaves is generally advantageous in arid environments. However, large leaves with less energy exchange efficiency seem advantageous in humid environments [70, 71].

Narrowing of leaves is an adaptive strategy to maintain a high perimeter within a finite area. Narrow leaves with optimal width have less photosynthetic productivity in terms of leaf area. However, the reduction in leaf area is compensated by the ability of elongated leaves to harvest water from fog [72] and tolerate strong shearing forces [73]. Leaf width and perimeter (lobiness) also describe the adaptive stability of plants by the boundary layer thickness of plant leaves. Lobed and narrow leaves usually have a thin leaf boundary layer than circular leaves [74, 75]. When the microclimatic boundary layer becomes thin, leaves can track the surrounding air through efficient cooling and heating by convection [75, 76]. Thus, lobed and narrow leaves are less vulnerable to heating and freezing during the day and night [77].

Since the form of geometric entropy of any 2-D object resembles $S_{GE}$, and here we consider the plant leaf as a 2-D object (see above), the parameter perimeter was taken as the leaf perimeter ($P$), and the smallest discretization length as the square root of the leaf area ($A$). $S_{GE}$ described here is a straightforward formula that evolved from an information perspective. $S_L$ increases with $P$ and decreases with $A$. The leaf area in the denominator of $S_L$ offsets the uncertainty of a large perimeter with entire leaf margins. It reveals the same structure as $LDI$. Leaves with a higher $P/A$ ratio have higher geometric entropy and complexity, ensuring more adaptive stability in changing environments (see above). Consequently, leaf geometry converges into fractal-like structures to accommodate excess leaf margin (within a finite area) by inducing waviness and lobiness along the edges. It is logical as fractal structures arise as a natural consequence of the most effective energy dissipation requirements [78, 79]. Natural fractals are an innovative adaptive strategy that minimizes energy/nutrient loss. Therefore, increasing geometric entropy opens discussions on the direct causal links between leaf stability/efficiency and complexity.

Leaves with similar morphological features have comparable $S_L$ values (Table 1). Narrow and pinnately compound leaves had a higher $P/A$ ratio and $S_L$ values. Consequently, narrow-simple leaves used in the study recorded higher $S_L$ than broad leaves with deep lobes. Irrespective of the size of the leaf laminar plane, $S_L$ is scalable by the $P/A$ ratio. Similar to the $S_L$ values, $D_{\Sigma S}$ increases with decreasing leaf width and increasing lobiness and pinnation.

Though $S_L$ and $D_{\Sigma S}$ were developed from two different notions, they are related. The strong linear relation between the $S_L$ and $D_{\Sigma S}$ (Fig 5) reinforces the validity of $S_L$. $S_L$ is developed from an information theory approach. It increases with increasing leaf perimeter and decreasing leaf area. Leaves with lobiness, dissection, narrow width, and high serrations have a high $P/A$ ratio and, thereby, high $S_L$. $D_{\Sigma S}$ is based on a fractal-thermodynamic system analogy that comprises discrete fractal dimensions of leaf parts: lamina, background, and edge. The principle of fractal analysis stems from the space-filling capacity of fractal parts (leaves), which describes the scaling and distribution of leaf parts from the surrounding space. $D_{\Sigma S}$ is lower for simple leaves and increases with thinning, lobiness, and pinnation of leaves. The ability to capture the plant leaves' dissection, lobiness, and serration features underlines the analogy between $S_L$ and $D_{\Sigma S}$.

Leaf morphological studies are crucial to plant taxonomy and systematics [1]. Univariate cluster analysis of $S_L$ distribution reveals the taxonomic relationship among the leaf shapes (Fig 6, Table 2). $S_L$ was classified into 5 clusters at a threshold similarity of 89%. Three-quarters of the species were clubbed into cluster 1 (red in Fig 6), and most had simple leaves (Fig 4). Simple leaves generally have comparable length and width as compared to compound leaves.

In such cases, the variations in $S_L$ due to the variations in perimeter and area (due to variations in length and width) will be less apparent in simple leaves than in compound leaves. This limits the $S_L$ of leaves in cluster 1 to be grouped together. Leaves in other clusters show significant $S_L$ variations (5.192–30.683). A consistent pattern of $S_L$ could not be attributed to plant leaves belonging to the same family. The leaves of plants from the same family exhibit various leaf shapes. However, a pattern seems to emerge at the genus level. Plant leaves of the species belonging to the same genus exhibit similar shapes and $S_L$ values. Therefore, we hope $S_L$ could stimulate plant biologists to explore its potential use in taxonomy.

$S_L$ is an inherent complexity measure that outperforms other complex geometric morphometrics. Devoid of statistical techniques, $S_L$ is free from time-consuming preprocessing techniques and can account for leaf shape regardless of the size of the leaves. It posits a prospective method to quantify the extent of variation in leaf shapes, especially by the influence of deep lobiness, serrations, and dissections on leaf perimeter. The ease of use and efficiency of $S_L$ will encourage plant biologists to draw more accurate inferences on leaf shape variations. Moreover, the theoretical maximum of $D_{\Sigma S}$ of extremely narrow leaves and the Peano curve-shaped leaves with a high *P/A* ratio [47] consolidates $S_L$. Since complex leaves (high *P/A* ratio) have more adaptive stability in changing environments [80, 81], $S_L$ can be considered a derived plant trait to describe leaf complexity and adaptive stability.

The evolution of leaf shapes is not by chance. It is an end-product of functional perfection [82]. The complexity bestowed in the leaves by evolution reflects directly on plants' physiological processes. The knowledge of complex leaf forms has a vast potential for understanding geometric information and its link with energy capture. The joint descriptions of information and energy will answer pertinent questions about nature's design procedures and energy-entropy tradeoffs. Our findings introduce an inclusive measure of leaf complexity, represented as geometric entropy. It demonstrates the utility and outperformance over conventional landmark-based and geometric morphometrics. $S_L$ is an objective plant trait that can be leveraged to measure leaf stability/efficiency. It will help in artificial leaf design studies to genetically engineer optimal leaf shapes to increase energy capture, carbon sequestration, and crop yields [83]. Similarly, since lobiness and serrations in leaves enhance the plant leaves thermal endurance and vapor dissipation, designing structures with evaporative protrusions inspired by leaf structures (Biomimetics) is being explored. Various model inspired by the leaf geometry reveals the correlation of evaporation rate with protrusion aspect ratio and breaks new ground for designing evaporation-assisted and passively enhanced thermal systems [84, 85]. Further, serrations in leaves enable a large area with optimized aerodynamic properties, viz. cooling and wind resistance, which finds potential applications in designing photovoltaic panels [86]. Geometric morphometric studies were complicated by the plasticity of features and the number of identifiable homologous points [87, 88]. Since scalable by the *P/A* ratio, $S_L$ can be considered a more practical measure that circumvents the objectivity constraint imposed by leaf plasticity. Further, geometric morphometric techniques mainly focus on the homologous features that are sensitive to the leaf size rather than leaf shape, which limits the comparison of leaves with disparate morphologies and thus cannot reliably discriminate leaf shapes at taxonomic levels. However, despite slight imperfections, $S_L$ posits a potential method for classifying leaf shapes at a genus level.

Leaf morphology is closely related to climate [89]. Biological specimens have been found useful in tracking the species' morphological changes caused by climate change [90, 91]. Since paleoclimate correlations focus on marginal serrations [53], leaf dissection (thereby $S_L$) can be utilized as a plausible index to understand the paleoclimate. Further, paleobotanical studies have confirmed an adaptive temporal shift towards narrow leaves [91], consolidating the structure and utility of $S_L$ in describing leaf adaptive stability/efficiency. Though several hypotheses

about the possible functions of dissected leaves have been discussed [52, 56, 92], detailed physiological insights describing possible changes in leaf shape in response to climate have not yet been revealed. More inferences can be revealed only by combining digital morphometrics with the paleobotanical collections. Incorporating $S_L$ as a morphological trait can help analyze the climatic relationship of leaf forms to understand further prospective applications, viz. models depicting the impacts of climate change scenarios on plants, reconstructing paleotemperature from paleobotanical leaf specimens, the evolutionary history of plants, and futuristic ecosystems [93]. Further, integrating $S_L$ into other leaf physiological measures such as photosynthesis, respiration, and evaporation can open pathways to understanding energy flow and interaction between plant leaves and their surroundings. Morphological complexity, expressed as $S_L$, will pave the way to understanding adaptive resilience at the species level. Resolving these gaps between information, entropy, and energy will allow advances to reveal ecosystem dynamics.

## Conclusion

We presented a mathematical framework to estimate the geometric entropy of plant leaves. The geometric entropy contains information on leaf geometry, represented by two physiologically important traits: leaf perimeter and leaf area. The geometric entropy of the leaf reveals the physical basis of its dissection index and accounts for the connection between complexity and adaptive stability in plants. However, leaves exhibit diverse leaf shapes at the family level. That limits $S_L$ to any taxonomic relationship at the family level. A consistent pattern of $S_L$ seems to emerge at the genus level. Further, we propose geometric entropy as a derived plant trait to discriminate leaf shapes and denote the adaptive stability of plants in rapidly changing environments. The relevance of this morphological trait needs to be tested to explore adaptive plant morphogenesis and obtain a clearer picture of function-based Eco-Devo studies.

## Supporting information

**S1 Data. Scanned images of plant leaves collected from Trivandrum, Kerala, India.** (ZIP)

## Acknowledgments

We thank Dr. Saji Gopinath, Vice-Chancellor, Kerala University of Digital Sciences, Innovation and Technology, India, for providing all necessary support to carry out the study. We further gratefully acknowledge the support of Ms. Aiswaria G R, Kerala University of Digital Sciences, Innovation, and Technology, for valuable comments that substantially improved the manuscript.

## Author Contributions

**Conceptualization:** Vishnu Muraleedharan, Jaishanker R.

**Data curation:** Vishnu Muraleedharan, Sajeev C. Rajan.

**Formal analysis:** Vishnu Muraleedharan.

**Investigation:** Vishnu Muraleedharan, Sajeev C. Rajan.

**Methodology:** Vishnu Muraleedharan, Jaishanker R.

**Supervision:** Jaishanker R.

**Validation:** Vishnu Muraleedharan.

**Visualization:** Vishnu Muraleedharan, Sajeev C. Rajan.

**Writing – original draft:** Vishnu Muraleedharan.

**Writing – review & editing:** Jaishanker R.

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
