## [Decision Letter · Decision Letter 0]

5 Jul 2023

PONE-D-23-07968Geometric Entropy of plant leaves: A measure of morphological complexityPLOS ONE

Dear Dr. Nair,

Thank you for submitting your manuscript to PLOS ONE. After careful consideration, we feel that it has merit but does not fully meet PLOS ONE’s publication criteria as it currently stands. Therefore, we invite you to submit a revised version of the manuscript that addresses the points raised during the review process.

We look forward to receiving your revised manuscript.

Kind regards,

Marcello Iriti, Ph.D.

Academic Editor

PLOS ONE

Journal Requirements:

Additional Editor Comments:

The reviewer is critical with your manuscript. We give you a chance to revise it.

Please reply satisfactorily to the reviewer, in particular explain and justify the benefit arising from these measures and their application(s).

Reviewers' comments:

Reviewer's Responses to Questions

**Comments to the Author**

1. Is the manuscript technically sound, and do the data support the conclusions?

Reviewer #1: Yes

2. Has the statistical analysis been performed appropriately and rigorously? 

Reviewer #1: N/A

3. Have the authors made all data underlying the findings in their manuscript fully available?

Reviewer #1: Yes

4. Is the manuscript presented in an intelligible fashion and written in standard English?

Reviewer #1: Yes

5. Review Comments to the Author

Reviewer #1: Te paper presents a mathematical framework to estimate the geometric entropy of plant leaves. The geometric

entropy contains information on leaf geometry, represented by two physiologically important traits, leaf

perimeter and leaf area. However the benefit and the application from the new measure are not well explaine and not justfied

6. PLOS authors have the option to publish the peer review history of their article (what does this mean?). If published, this will include your full peer review and any attached files.

Reviewer #1: No

---

## [Author Response · Author response to Decision Letter 0]

13 Jul 2023

REVIEWER REPORT(S):

Kindly note: Our response to the reviewer’s comments is solely based on the email received. It did not have any attachments.

COMMENTS TO THE AUTHOR(S)

Reviewer: 1

The paper presents a mathematical framework to estimate the geometric entropy of plant leaves. The geometric entropy contains information on leaf geometry, represented by two physiologically important traits, leaf perimeter and leaf area. However the benefit and the application from the new measure are not well explaine and not justified

Thank you very much for taking the time to review our manuscript and for your insightful comments.

We added four more plant leaves into the dataset and accommodated them in a single plate (Fig 4). We reanalysed the work with 112 leaves, which did not affect/alter the result or conclusion.

A detailed explanation of the benefits and application of geometric entropy measures with justification by references was included in the discussion (Page 31. line 433-467).

A section justifying the assumptions and scientific derivation of the geometric entropy of plant leaves (SL) from geometric entropy of circular objects is replaced from ‘Discussion’ into ‘Geometric entropy of plant leaf’ (Page 11. line 192-203).

All the changes were incorporated in the manuscript under tracked changes.

Additional Editor Comments:

The reviewer is critical with your manuscript. We give you a chance to revise it.

Please reply satisfactorily to the reviewer, in particular explain and justify the benefit arising from these measures and their application(s).

Thank you for offering the chance to revise the manuscript.

We have accounted for the suggestions by the reviewer. A detailed explanation of the benefits and application of geometric entropy measures with justification by references was included in the discussion (Page 31. line 433-467).

A section justifying the assumptions and scientific derivation of the geometric entropy of plant leaves (SL) from geometric entropy of circular objects is replaced from ‘Discussion’ into ‘Geometric entropy of plant leaf’ (Page 11. line 192-203).

Further, we reanalyzed the work by including four additional plant leaves, which did not affect/alter the result or conclusion. All the changes were incorporated in the manuscript under tracked changes.

---

## [Decision Letter · Decision Letter 1]

3 Oct 2023

PONE-D-23-07968R1Geometric entropy of plant leaves: A measure of morphological complexityPLOS ONE

Dear Dr. Nair,

Thank you for submitting your manuscript to PLOS ONE. After careful consideration, we feel that it has merit but does not fully meet PLOS ONE’s publication criteria as it currently stands. Therefore, we invite you to submit a revised version of the manuscript that addresses the points raised during the review process.

We look forward to receiving your revised manuscript.

Kind regards,

Marcello Iriti, Ph.D.

Academic Editor

PLOS ONE

Journal Requirements:

Additional Editor Comments:

please see the reviewer's attached file

Reviewers' comments:

Reviewer's Responses to Questions

**Comments to the Author**

1. If the authors have adequately addressed your comments raised in a previous round of review and you feel that this manuscript is now acceptable for publication, you may indicate that here to bypass the “Comments to the Author” section, enter your conflict of interest statement in the “Confidential to Editor” section, and submit your "Accept" recommendation.

Reviewer #2: All comments have been addressed

2. Is the manuscript technically sound, and do the data support the conclusions?

Reviewer #2: Yes

3. Has the statistical analysis been performed appropriately and rigorously? 

Reviewer #2: Yes

4. Have the authors made all data underlying the findings in their manuscript fully available?

Reviewer #2: Yes

5. Is the manuscript presented in an intelligible fashion and written in standard English?

Reviewer #2: Yes

6. Review Comments to the Author

Reviewer #2: The work is simple, but relevant as it can provide a parameter for understanding physiological processes that involve the leaf surface, such as gas exchange and heat.

7. PLOS authors have the option to publish the peer review history of their article (what does this mean?). If published, this will include your full peer review and any attached files.

Reviewer #2: **Yes: **Marcio Paulo Pereira

---

## [Author Response · Author response to Decision Letter 1]

8 Oct 2023

REVIEWER REPORT(S):

COMMENTS TO THE AUTHOR(S)

Journal Requirements:

We have reviewed all the references. The reference list is complete and correct. We have not cited any papers that have been retracted.

Reviewer: 2

All comments have been addressed.

Thank you very much for taking the time to review our manuscript and your positive feedback.

The work is simple, but relevant as it can provide a parameter for understanding physiological processes that involve the leaf surface, such as gas exchange and heat.

Thank you very much for taking the time to review our manuscript and for your insightful comments. We have added a few supporting lines in the discussion (page: 32, line: 444-451) to improve the flow and clarity.

All the changes were incorporated in the manuscript under tracked changes.

Additional Editor Comments:

Please see the reviewer's attached file.

We have considered the reviewer's attached file and accounted for the suggestions by the reviewer (page:20-21, line: 249-252).

All the changes were incorporated in the manuscript under tracked changes.

---

## [Editor Report · Decision Letter 2]

17 Oct 2023

Geometric entropy of plant leaves: A measure of morphological complexity

PONE-D-23-07968R2

Dear Dr. Nair,

We’re pleased to inform you that your manuscript has been judged scientifically suitable for publication and will be formally accepted for publication once it meets all outstanding technical requirements.

Kind regards,

Marcello Iriti, Ph.D.

Academic Editor

PLOS ONE
---

## [Editor Report · Acceptance letter]

20 Dec 2023

PONE-D-23-07968R2 

PLOS ONE

Dear Dr. R, 

I'm pleased to inform you that your manuscript has been deemed suitable for publication in PLOS ONE. Congratulations! Your manuscript is now being handed over to our production team.

Kind regards, 

on behalf of

Prof. Marcello Iriti 

Academic Editor

PLOS ONE